

# SLiMEnrich: computational assessment of protein–protein interaction data as a source of domain-motif interactions

Sobia Idrees, Åsa Pérez-Bercoff and Richard J. Edwards

School of Biotechnology and Biomolecular Sciences, University of New South Wales, Sydney, NSW, Australia

## ABSTRACT

Many important cellular processes involve protein–protein interactions (PPIs) mediated by a Short Linear Motif (SLiM) in one protein interacting with a globular domain in another. Despite their significance, these domain-motif interactions (DMIs) are typically low affinity, which makes them challenging to identify by classical experimental approaches, such as affinity pulldown mass spectrometry (AP-MS) and yeast two-hybrid (Y2H). DMIs are generally underrepresented in PPI networks as a result. A number of computational methods now exist to predict SLiMs and/or DMIs from experimental interaction data but it is yet to be established how effective different PPI detection methods are for capturing these low affinity SLiM-mediated interactions. Here, we introduce a new computational pipeline (SLiMEnrich) to assess how well a given source of PPI data captures DMIs and thus, by inference, how useful that data should be for SLiM discovery. SLiMEnrich interrogates a PPI network for pairs of interacting proteins in which the first protein is known or predicted to interact with the second protein via a DMI. Permutation tests compare the number of known/predicted DMIs to the expected distribution if the two sets of proteins are randomly associated. This provides an estimate of DMI enrichment within the data and the false positive rate for individual DMIs. As a case study, we detect significant DMI enrichment in a high-throughput Y2H human PPI study. SLiMEnrich analysis supports Y2H data as a source of DMIs and highlights the high false positive rates associated with naïve DMI prediction. SLiMEnrich is available as an R Shiny app. The code is open source and available via a GNU GPL v3 license at: https://github.com/slimsuite/SLiMEnrich. A web server is available at: http://shiny.slimsuite.unsw.edu.au/SLiMEnrich/.

# INTRODUCTION

Proteins interact with their partners through two main classes of functional modules: globular domains and Short Linear Motifs (SLiMs) (*Bhattacharyya et al., 2006*). SLiMs are short protein regions (typically 3–10 amino acids long) with a small number of key residues that mediate domain-motif interactions (DMIs) with the globular domain of a protein–protein interaction (PPI) partner (*Davey et al., 2012*). These DMIs underpin critical cellular functions, including cell cycle regulation, cell compartment targeting, post-translational modification, protein degradation, and signal transduction

Corresponding author
Richard J. Edwards,
richard.edwards@unsw.edu.au

(*Van Roey et al., 2014*). Knowledge of DMIs can provide molecular details of cellular processes and thus it is important to discover SLiMs and link them to their domain partners (*Davey et al., 2012*; *Neduva & Russell, 2005*). Despite this, only a small fraction of the likely range of SLiMs, and the DMIs they mediate, have been identified (*Tompa et al., 2014*) and curated in resources such as the Eukaryotic Linear Motif (ELM) resource (*Gouw et al., 2018*), Linear Motif mediated Protein Interaction Database (LMPID) (*Sarkar, Jana & Saha, 2015*), interActions of moDular domAiNs (ADAN) (*Encinar et al., 2009*), and the database of three-dimensional interacting domains (3did) (*Mosca et al., 2014*). SLiM-mediated interactions are typically low affinity (*Davey et al., 2012*) and are thus vulnerable to being overlooked by classical PPI detection methods, such as affinity pulldown mass spectrometry (AP-MS) and yeast two-hybrid (Y2H), where high stringencies are typically employed to reduce false positive interactions. Early analyses of high throughput data revealed that known SLiM-mediated interactions account for less than 1% of interactions (*Neduva & Russell, 2006*). This was used as evidence that many more SLiMs and DMI are yet to be discovered, but also raises concerns that these methods are depleted for DMIs.

A range of computational tools now exist for the two main tasks in SLiM prediction: (1) identifying functional instances of known motifs, and (2) *de novo* prediction of new SLiM classes (*Edwards & Palopoli, 2015*). In principle, the task of interrogating a protein sequence for known motif patterns is quite simple. Motif definitions are available from ELM (*Gouw et al., 2018*; *Gouw et al., 2017*) and PROSITE (*Hulo et al., 2006*), and various tools exist for searching proteins for these patterns or resource-specific motif definitions (*Edwards & Palopoli, 2015*). Other tools, like Minimotif Miner (MnM) (*Lyon et al., 2018*), will search sequences for similarity to known SLiMs or post-translational modifications (PTMs), but do not make motif definitions or tools available for proteome-scale searches. The short and degenerate nature of most SLiMs hampers the usefulness of predictions due to the high possibility of false positive results. This is particularly true for SLiMs with very few known occurrences, which will lack the data required for detailed modelling. It is therefore important to improve the specificity of predictions by incorporating contextual information such as evolutionary conservation and/or protein structure (*Krystkowiak & Davey, 2017*; *Mi et al., 2012*), or knowledge of interaction partners containing relevant SLiM recognition domains (e.g., *Encinar et al., 2009*; *Kelil, Levy & Michnick, 2016*; *Luck et al., 2011*; *Weatheritt et al., 2012*).

The *de novo* prediction of SLiMs is inherently more challenging and relies on assembling sets of proteins that share a SLiM. The most widespread approach is to mine PPI data to identify sets of proteins that interact with a common partner (e.g., *Edwards et al., 2012*; *Lieber, Elemento & Tavazoie, 2010*; *Neduva et al., 2005*). The success of prediction methods is highly dependent on the signal to noise ratio in these data, in terms of the proportion of proteins likely to contain the SLiM (*Edwards et al., 2012*; *Edwards & Palopoli, 2015*). Before attempting SLiM discovery, it is therefore useful to know how well the input PPI data is capturing SLiM-mediated interactions. Different experimental parameters will influence how depleted the recovered interactions are for DMIs, and so this assessment is also useful for experimentalists when establishing an appropriate stringency threshold.

Here, we introduce a new computational pipeline (SLiMEnrich) that assesses how well PPI data are capturing DMIs and thus, by inference, how useful that data should be for SLiM discovery. SLiMEnrich evaluates DMI enrichment through permutation tests and reports the probability of randomly recovering as many interacting domain-motif pairs as are found in the real PPI data. SLiMEnrich can use known SLiM-mediated interactions for high stringency analysis, or incorporate DMI predictions by using SLiM predictions and/or known SLiM-domain interactions to expand the number of plausible DMIs in the data. Identified/predicted DMIs are returned, along with an estimated false discovery rate based on the mean number of random DMIs generated from the data. Whilst not their primary purpose, SLiMEnrich metrics can also be used to assess SLiM and/or DMI prediction strategies when applied to PPI data that is already known to contain DMIs. SLiMEnrich is therefore of potential use for both DMI prediction and assessment of PPI data. SLiMEnrich has been developed in R and implemented in Shiny to provide easy, user-friendly operation.

## MATERIALS AND METHODS

### Algorithm

An overview of the SLiMEnrich pipeline is shown in Fig. 1. SLiMEnrich uses (known or predicted) SLiM occurrences, domain composition, and known SLiM interactions at the protein or domain level. These are combined to predict SLiM-mediated DMIs within pairwise PPI data supplied by the user. Input data is combined by matching protein, SLiM and Domain IDs from the input data, providing a flexible framework for analysis. PPI data is treated asymmetrically, with specified sets of putative motif- and domain-containing proteins, known as "mProteins" and "dProteins", respectively. First, SLiMEnrich identifies all possible known/predicted DMI links between mProteins and dProteins in the PPI data (Fig. 1). DMI mapping can be performed using a number of different strategies depending on the desired balance of quality versus quantity of DMI (Figs. 1, 2). At one extreme, analysis can be restricted to mProtein–dProtein pairs known to interact via a DMI (Fig. 2, top left). At the other extreme, mProteins with predicted SLiMs can be linked to any dProteins containing a domain known to interact with that SLiM (Fig. 2, bottom right). This set of "potential DMIs" represents the overall pool of possible DMIs given the input data and mapping strategy.

Next, SLiMEnrich extracts "predicted DMIs" by identifying the subset of potential DMIs that are found in the PPI data, e.g., observed PPI pairs where the mProtein is known or predicted to interact with the dProtein according to the DMI strategy employed. Finally, SLiMEnrich estimates how well the PPI data is capturing DMIs by comparing the observed DMI predictions to a background distribution of expected DMIs when proteins are randomly assigned interaction partners. For this, the input PPI data is shuffled to generate 1,000 random PPI datasets where each protein maintains the same number of interacting partners but the connections are randomly assigned. This is performed by first reducing PPI data to asymmetrical non-redundant protein pairs and then randomly shuffling the dProtein column whilst avoiding the introduction of redundant random PPI pairs. The
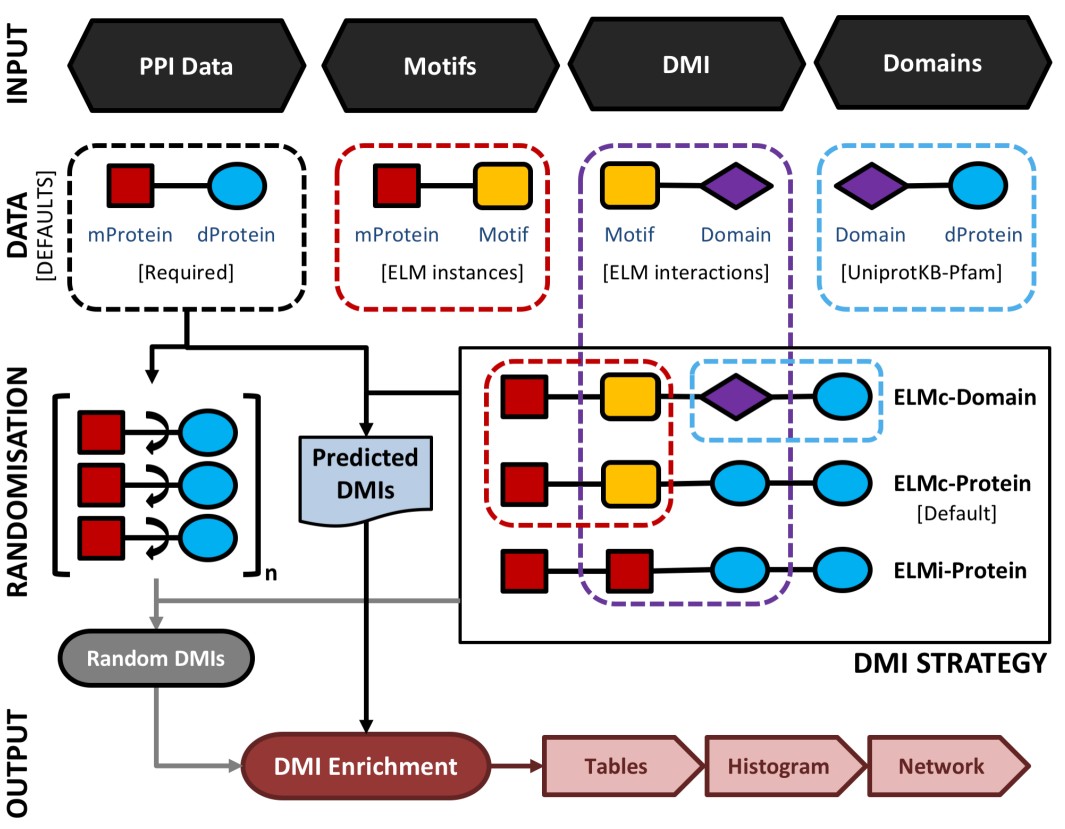

**Figure 1  A schematic representation of the main SLiMEnrich pipeline.** SLiMEnrich takes four input files: 1. PPI data provided by the user as a set of pairwise putative motif-containing proteins ("mProteins") and their domain-containing interaction partners ("dProteins"); 2. a file providing known or predicted motif occurrences within the mProtein sequences (by default, known ELM instances are used); 3. a DMI file defining Motif-Domain interactions, relating to the DMI Strategy employed (by default, known ELM interactions are used); 4. a file that links dProteins to their domain composition (by default, human Pfam domains from UniprotKB are used). Input data is combined to establish the complete set of known/predicted "potential DMI" dependent on the DMI strategy selected (see Fig. 2 and text for details): ELMi-Protein –for highest stringency, the DMI file directly links mProteins to known dProtein DMI partners (motifs and domains input not used); ELMc-Protein –for medium stringency, the DMI file links Motif classes to known dProtein DMI partners (Domains input not used); ELMc-Domain –for lowest stringency, the DMI file links Motif classes to known interacting domains. Potential DMIs are then mapped on to the input PPI to identify the "Predicted DMIs" in the real data. PPI data is randomised (shuffled) 1,000 times and re-mapped to potential DMIs to determine the background distribution of predicted DMIs in the case of random association (see text for details). Finally, the "Random DMI" distribution is compared to the observed "Predicted DMIs" to determine DMI enrichment in the data. Results are output in the form of a tables, a histogram of the Random DMI distribution with the observed count and empirical $P$-value marked, and an interactive network of the known/predicted DMIs found in the PPI data.

random PPI datasets are then mapped onto the potential DMIs in the same fashion as the real data. Enrichment is calculated as an empirical $P$-value corresponding to the probability of seeing at least as many DMIs in random PPI data (Fig. 3). A False Discovery Rate (FDR) for individual DMIs is also estimated as the proportion of the predicted DMIs explained on average by random associations, using the mean random DMI distribution capped at the observed value.
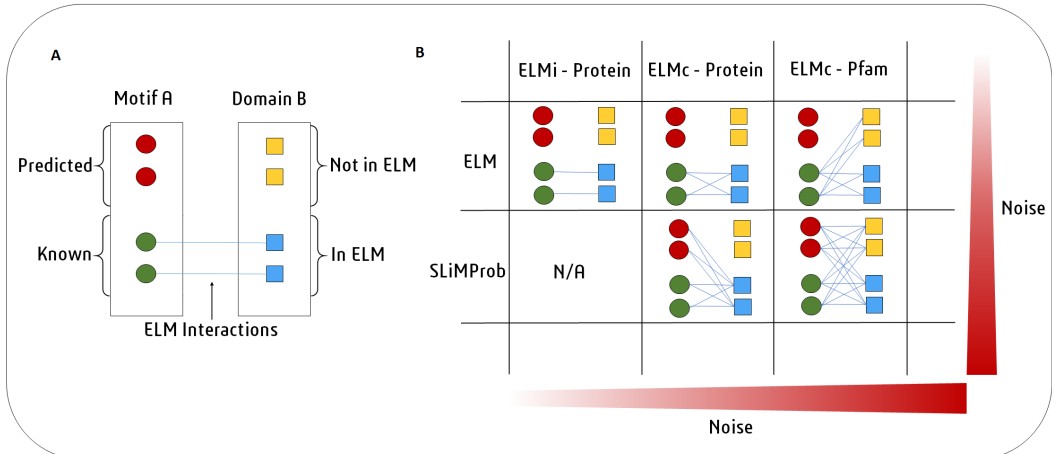

**Figure 2  SLiMEnrich uses known DMI from the ELM database to identify known DMIs or predict DMIs within the supplied PPI data.** (A) In this example, Motif A is known to interact with Domain B. Motif A has two known occurrences in the data (green circles) and two predicted occurrences (red circles). Domain B is present in four proteins (squares). ELM has two annotated interactions between proteins with Motif A and proteins with Domain B (blue). (B) In the simplest and purest strategy, only known ELM interactions (ELMi - Protein) are used to assess enrichment. For small PPI datasets it might be necessary to increase the number of predicted DMI. This can be done in two ways. ELM: known motif occurrences (green circles) can be connected to all proteins known to interact with that ELM class (blue squares, ELMc - Protein), or connected to all proteins containing a domain that interacts with that ELM class (all squares, ELMc -Pfam). SLiMProb: to increase the number of DMI further, known ELM occurrences can be replaced with SLiM predictions (all circles).

## Requirements and implementation
### Inputs

SLiMEnrich requires a delimited pairwise PPI file as input. By default, known ELM instances (ELMi) (*Gouw et al., 2018*) will be used to define the motif composition of mProteins. This file can be replaced by a SLiM prediction file (generated by e.g., SLiMProb *Edwards & Palopoli, 2015*), which has predicted SLiMs for the mProteins in the PPI file. DMIs can be predicted by one of three strategies (Fig. 2). By default, the DMI file links ELM classes (ELMc) directly to dProteins using known ELM binding partners (*Gouw et al., 2018*). For more stringent analysis, these binding partners can be linked directly to specific ELM-containing proteins, in which case the DMI file links mProteins and dProteins, and the motif occurrence file is ignored (Fig. 1). For more relaxed/flexible analysis, the DMI file will link motifs to binding domains, which are then linked to dProteins via a domain composition file. By default, SLiMEnrich uses Pfam domains (*Finn et al., 2016*) for reviewed human Uniprot proteins (*UniProt Consortium, 2017*) and links them to ELM-binding domains (*Gouw et al., 2018*). If alternative data sources are used, users should also provide a file of protein-domain links for the dProteins in the PPI file, and/or a motif-domain file that defines the known domain-motif interactions. Note that this can be used to interrogate PPI data for enrichment of any interaction type. For example, two protein-domain files could be linked through known domain-domain interactions. Alternatively, the ELMi-Protein DMI strategy enables the enrichment analysis of any set

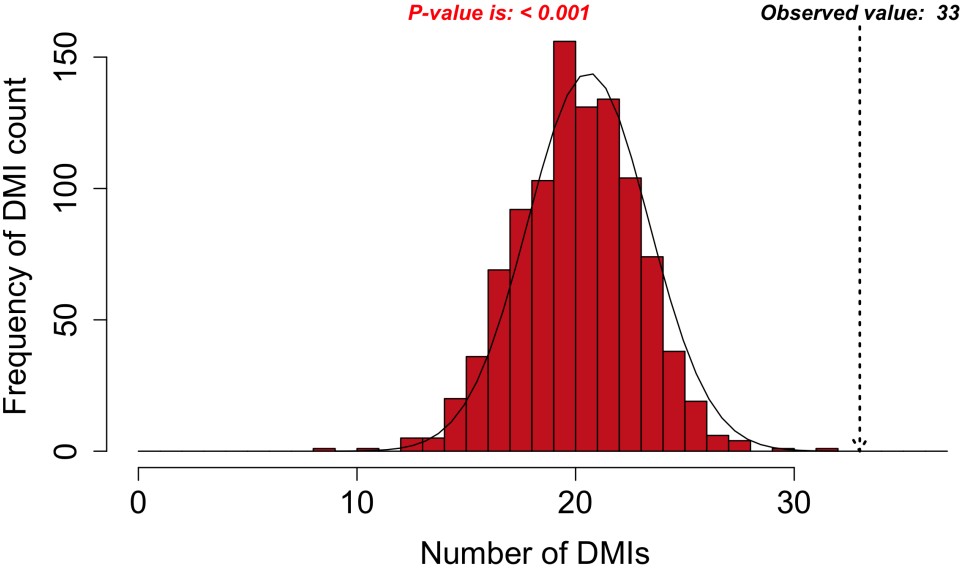

**Figure 3** **DMI enrichment histogram for SLiMEnrich example data.** Histogram of DMI enrichment in example data for Adenoviridae proteins and their human interactors (see text for details) from the SLiMEnrich app, using the most permissive ELMc-Domain DMI strategy and SLiMProb motif predictions. Frequency bars indicate the number of randomised PPI datasets returning a given number of predicted DMIs. The dotted arrow indicates the observed number of predicted DMIs in the real data.

of PPIs, allowing SLiMEnrich to examine overlaps between PPI datasets. Default fields for user files ("mProtein", "dProtein", "Motif", "Domain") are show in Fig. 1, and can be set to custom values in the SLiMEnrich App.

### Example data
SLiMEnrich comes with example data of Adenoviridae proteins and their human interactors downloaded from the PHISTO database (2017-07-26) (*Durmus Tekir et al., 2013*). ELM (downloaded 2018-07-17) (*Dinkel et al., 2016*) regular expression matches in the viral proteins were predicted using SLiMProb v2.5.0 (*Edwards & Palopoli, 2015*) with disorder masking. A table of ELM-binding Pfam domains was downloaded from ELM (2018-07-17) (*Dinkel et al., 2016*). Pfam domains for human proteins were extracted from Uniprot (downloaded 2017-03-08) (*UniProt Consortium, 2017*).

### Outputs
The primary output of SLiMEnrich is the observed number of known/predicted DMIs compared to the distribution from the randomised PPI data (Fig. 3). SLiMEnrich also provides tables of both "potential DMIs" and "predicted DMIs" (Fig. 1, see Algorithm for details), summary plots of predicted DMI numbers and an interactive DMI network (Fig. 4). Together, these enable the user to explore the data for proteins, SLiMs and/or domains that might be biasing results. This can be seen with the example Adenoviridae analysis, where the Pkinase domain (PF00069) mediates a large proportion of the predicted DMIs via multiple modification ELMs (Fig. 4), which will inflate the probability of DMIs
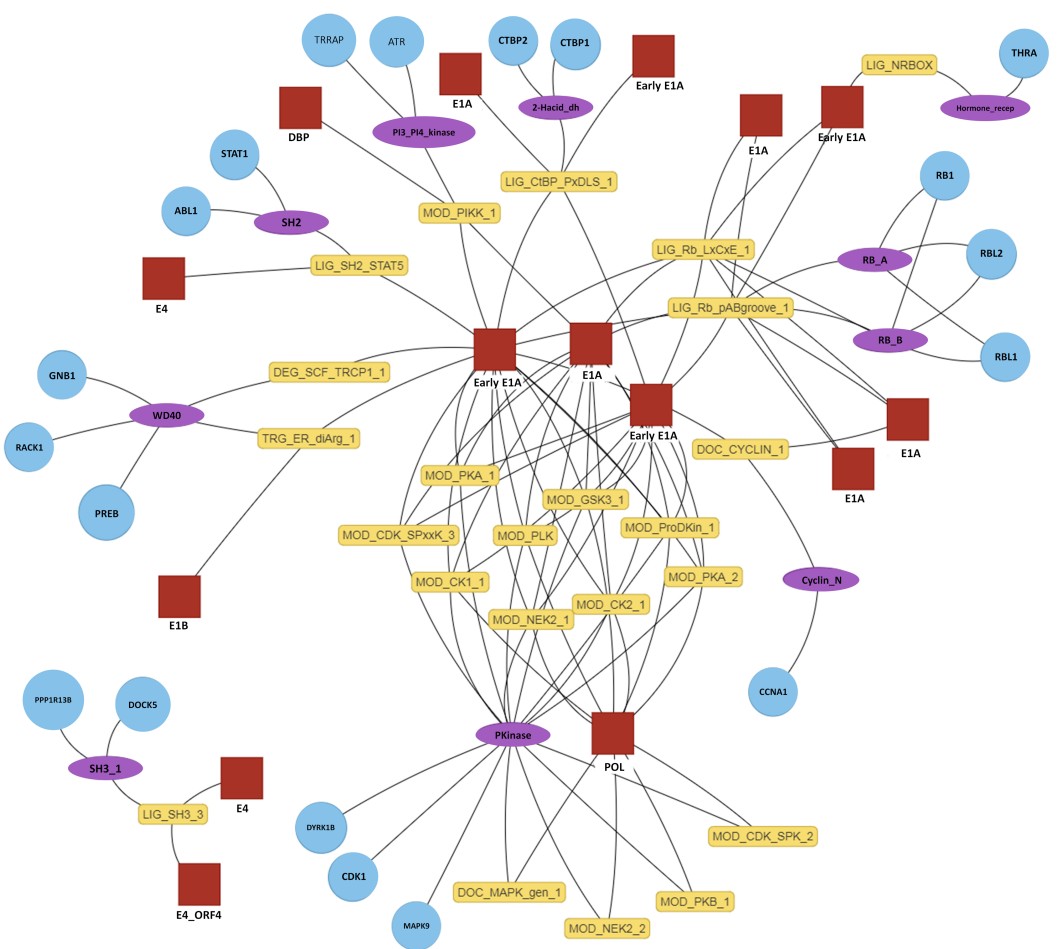

**Figure 4  Interactive predicted DMI network for example data.** Predicted DMIs for example Adenoviridae proteins and their human interactors, using the most relaxed strategy (predicted SLiMs connected via domains, see text for details). Several layout options are provided and nodes can be manually positioned. The protein, domain and motif identifiers used in the network are determined by the user input. Using default data, these will be UniprotKB, Pfam and ELM identifiers. For this example, UniprotKB identifiers have been mapped onto HGNC gene symbols and Pfam identifiers onto Pfam domain names. Red square, motif-containing protein ("mProtein"); Yellow box, motif; Purple ellipse, domain; Blue circle, domain-containing protein ("dProtein").

in the random PPI data. Tables can be downloaded as comma-separated text files. The summary plots, enrichment histogram and DMI network can be downloaded as PNG files.

### Implementation
SLiMEnrich is a standalone application written entirely in R. It is platform independent and can be launched locally from any R environment (e.g., RStudio). SLiMEnrich takes advantage of the reactive programming feature of Shiny to cache computational steps to avoid unnecessary computing during an interactive session. The code is open source and available via a GNU GPL v3 license at: https://github.com/slimsuite/SLiMEnrich. SLiMEnrich is also implemented as a Shiny webserver at: http://shiny.slimsuite.unsw.

edu.au/SLiMEnrich/. Additional details can be found at: https://github.com/slimsuite/SLiMEnrich/wiki.

## Case study: Domain-motif resolved yeast-two-hybrid human interactome

Pairwise human PPIs were extracted from a high-throughput human Y2H study that detected ∼14,000 binary interactions (*Rolland et al., 2014*) and converted into a non-redundant, symmetrical PPI dataset of 26,166 mProtein–dProtein PPIs (i.e., with each PPI pair present as P1-P2 *and* P2–P1), restricted to reviewed Uniprot proteins. Protein sequences were downloaded from Uniprot (2017-03-01). A list of ELMs and their domain partners was retrieved from the ELM database (2018-07-17) (*Dinkel et al., 2016*). ELM occurrences in the human proteins were predicted by SLiMProb v2.5.0 (*Edwards & Palopoli, 2015*) with disorder masking (IUPred (*Dosztanyi et al., 2005*), cut-off 0.2 (*Edwards, Davey & Shields, 2007*)) to restrict analysis to low stringency predicted disordered regions. Pfam domains were parsed from Uniprot entries using SLiMBench (*Palopoli, Lythgow & Edwards, 2015*). Splice isoforms for all data were mapped onto their parent Uniprot identifier. SLiMEnrich was used to map known and predicted DMIs onto the Y2H dataset using five strategies of decreasing stringency: (1) known ELM PPIs only, (2) known ELM instances mapped onto proteins known to interact with the ELM class, (3) known ELM instances mapped onto Pfam domains known to interact with the ELM class, (4) SLiMProb predictions mapped onto proteins known to interact with the ELM class, (5) SLiMProb predictions mapped onto Pfam domains known to interact with the ELM class (Fig. 2).

## Simulation of poor quality SLiM predictions

SLiMEnrich is not a DMI prediction tool *per se* and should not require completely accurate SLiM occurrence data to identify enrichment indicative of PPI data that captures DMIs. To investigate the impact of noisy SLiM prediction data, we replaced increasing proportions (25%, 50%, 75% and 100%) of the known ELM instances (2018-07-17) (*Dinkel et al., 2016*) with random occurrences and repeated analysis of the Y2H interactome case study. This was performed by replacing different proportions of the ELM proteins (i.e., proteins containing a known ELM) with a protein randomly selected from reviewed human Uniprot proteins (*UniProt Consortium, 2017*). For direct comparison, the distribution of normalised predicted DMIs, *D*, was calculated as follows:

$$D = \frac{O - R}{\bar{R}},$$

where $O$ is the observed predicted DMI count, $R$ is the distribution of random predicted DMIs, and $\bar{R}$ is the mean random predicted DMI count.

## RESULTS

### Case study: Domain-motif resolved yeast-two-hybrid human interactome

SLiMEnrich analysis revealed the case study Y2H data to be enriched for DMIs under all DMI prediction strategies (Fig. 5, Table 1). Restricting analysis to known DMIs identified
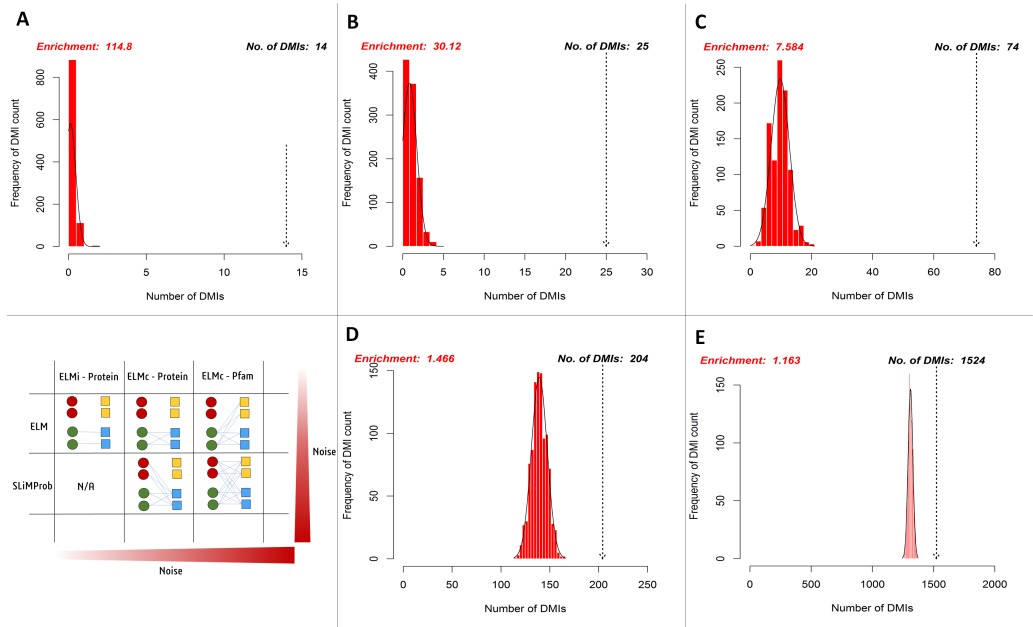

**Figure 5   Enrichment statistics and histogram of expected random DMI counts in human Y2H case study data using known and predicted ELM instances.** Frequency bars indicate the number of randomised PPI datasets returning a given number of predicted DMIs. The dotted arrow indicates the observed number of known or predicted DMIs in the Y2H data (see text for details). DMI prediction strategies match those in Fig. 2 (see text for details): (A) known ELM occurrences connected to interacting proteins; (B) known ELM occurrences mapped to proteins known to interact with that motif class; (C) known ELM occurrences mapped to proteins containing a domain known to interact with that motif class; (D) SLiM predictions mapped to proteins known to interact with that motif class; (E) SLiM predictions mapped to proteins containing a domain known to interact with that motif class.

fourteen in the Y2H data, which represented a more than 100-fold enrichment over the random expectation (mean 0.122). Including DMIs where a dProtein was known to interact with the ELM class (Fig. 2, ELMc-Protein), almost doubled the number of predicted DMIs but with nearer six times more random DMIs on average, reducing the enrichment over three-fold. Including DMIs where a dProtein contained Pfam domain known to interact with the ELM class (Fig. 2, ELM) dramatically increased the numbers of both predicted and random DMIs, with a corresponding drop in enrichment. Using SLiMProb predictions in place of known ELMs (Fig. 2, SLiMProb) similarly increased both predicted and random DMIs, decreasing enrichment. In all cases, none of the 1,000 randomised datasets matched or exceeded the observed number of predicted DMI, making the enrichment strongly significant ($P < 0.001$).

## Simulation of poor quality SLiM predictions

To directly compare the effects of replacing real ELM-containing proteins with random human proteins in different proportions (25%, 50%, 75%, 100%), the distribution of normalised predicted DMI, $D$, in the Y2H data was compared for each dataset (Fig. 6). $D$ is the distribution of expected true positive predicted DMIs, normalised to units of mean random predicted DMIs, i.e., $D = 1$ is equivalent to FDR = 50%; enrichment is $1 +$ mean $D$.

**Table 1  SLiMEnrich analysis of Y2H case study using different DMI prediction strategies.**

| SLiMEnrich strategy | Known: ELMi-Protein | Known: ELMc-Protein | Known: ELMc-Domain | SLiMProb: ELMc-Protein | SLiMProb: ELMc-Domain |
|---|---|---|---|---|---|
| Potential DMI (NR) | 62 | 164 | 6,314 | 39,572 | 969,380 |
| Predicted DMI (NR) | 14[b] | 25 | 74 | 204 | 1,524 |
| Mean Random DMI (3 s.f.) | 0.122 | 0.830 | 9.76 | 139 | 1,310 |
| *p*-value | <0.001 | <0.001 | <0.001 | <0.001 | <0.001 |
| Enrichment (3 s.f.) | 115 | 30.1 | 7.58 | 1.47 | 1.16 |
| FDR (4 d.p.) | 0.0087 | 0.0332 | 0.1319 | 0.6820 | 0.8602 |
| Unique mProteins[a] | 13 | 22 | 52 | 175 | 768 |
| Unique ELM classes[a] | N/A | 16 | 40 | 35 | 128 |
| Unique Pfam domains[a] | N/A | N/A | 30 | N/A | 51 |
| Unique dProteins[a] | 10 | 17 | 53 | 36 | 366 |

**Notes.**
[a] Unique counts correspond to Predicted DMI.
[b] Known DMI from ELM database.

The more permissive domain-based DMI prediction strategy (Fig. 2, top right) was used, as the numbers of predicted DMIs for more stringent strategies were very small (Table 1) and this strategy still showed strong (7.6×) DMI enrichment in the data (Fig. 5). Despite the decline in enrichment scores with increasing proportions of random motif occurrences, enrichment remained significant even when 75% of the real data was replaced (Fig. 6).

## DISCUSSION

Using PPI data for SLiM discovery faces something of a contradiction. Due to their scale, data from high throughput PPI detection studies are where the novel interactions are most likely to be found. However, high stringency filters are often applied to high throughput methods to increase confidence in individual interactions, with the concomitant concern that low affinity DMIs will be lost as a consequence. The primary purpose of SLiMEnrich is to address this concern by assessing how well a given PPI dataset is capturing DMIs. Where PPI datasets are large, this assessment can be restricted to a high-quality set of known DMIs. Where the number of known DMIs in the data becomes prohibitively small, predicted DMIs can supplement or replace the known DMIs.

A detailed analysis of different PPI data sources is the subject of future study and beyond the scope of this paper. Here, we present a case study to illustrate the use of SLiMEnrich to analyse the DMI enrichment in a single PPI dataset. We have applied five different DMI identification/prediction strategies (Fig. 2) to a high-throughput Y2H human PPI study (*Rolland et al., 2014*) (Table 1, Fig. 5). On face value, the ability of the Y2H PPI data to capture known DMIs might be considered disappointing. Only 14 of the 590 known human DMI protein pairs in ELM (2.37%) were found in the 26,166 PPI considered. This is consistent with earlier analyses that have highlighted the rarity of known SLiM-mediated interactions in high throughput PPI data (*Neduva & Russell, 2006*). However, even this modest number reflects a massive enrichment (approx. 115-fold) over the expected number of known DMIs to occur in the PPI data by chance. Whilst we cannot rule out

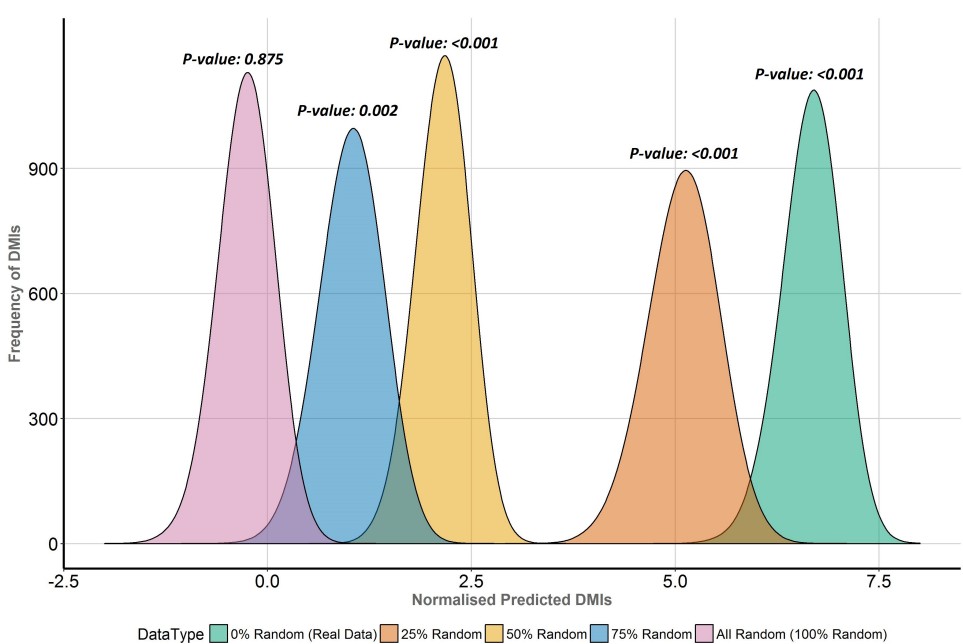

**Figure 6** **Enrichment analysis of known DMIs in human Y2H case study data with increasing proportions of random motif instances.** SLiMEnrich results for known ELMs in the human Y2H case study data mapped using the ELMc-Domain strategy, converted into the normalised number of predicted real DMIs (see text for details). Higher normalised predicted DMI counts indicate greater DMI enrichment, with zero marking no enrichment over random. Green is the real data using all known true positive ELM instances. The other curves (right to left) represent distributions for four randomised datasets where increasing proportions (25%, 50%, 75% and 100%) of ELM proteins were replaced with random human proteins.

unexpected confounding factors, such as additional high affinity interactions between pairs of proteins that also share a DMI, this implies that the low absolute numbers are due to the small number of known DMIs rather than the inability of Y2H methods to detect DMIs. Considered analysis has estimated that the human proteome has in the order of 100,000 SLiMs involved in DMIs (ignoring post-translational modifications) (*Tompa et al., 2014*), which is orders of magnitude greater than the known DMIs in ELM (*Gouw et al., 2018*). Overall, SLiMEnrich results indicate that these data are indeed capturing real SLiM-mediated interactions and are therefore suitable for *de novo* SLiM prediction. This, in turn, increases confidence in previous large scale SLiM predictions (*Edwards et al., 2012*; *Lieber, Elemento & Tavazoie, 2010*; *Neduva et al., 2005*); these often rely on rediscovery of known motifs as validation, which could be biased by incorporation of literature-based high confidence DMIs in the PPI data.

Employing a less stringent DMI identification strategy predictably boosted the numbers of predicted DMIs and continued to reveal significant enrichment in the Y2H data despite the possible incorporation of possible false positive SLiM and/or DMI predictions (Table 1, Fig. 5). As expected, the enrichment decreased as the noisiness of the data increased, although the enrichment remained highly significant. This was supported by analysis where real ELM-containing proteins were replaced with random human proteins to simulate noise

(Fig. 6). Taken together, these results indicate a degree of robustness of the SLiMEnrich approach to the quality of the SLiM data. However, they also highlight a lack of robustness in the individual DMI predictions. For the purest known DMI analysis (linking known ELM instances to known ELM-interacting *proteins*), most randomised datasets did not return a single DMI. It is therefore highly likely that the 11 additional DMIs discovered by the ELMc-Protein strategy are real DMIs. The cost is that the low numbers might affect the accuracy with which the mean random DMI count, and thus enrichment, can be calculated. Relaxing the strategy to use SLiMProb predictions and/or allow DMI predictions based on interactions between ELM classes and Pfam domain classes, substantially increased the numbers of predicted DMIs but dramatically reduced the observed enrichment for both known and predicted SLiM occurrences. Using predicted SLiMs, it should be noted that the estimated false positive rate for individual DMI predictions is very high (FDR = 0.86 when linking predicted SLiMs via ELM-binding Pfam domains). This highlights the need for caution when interpreting naïve large-scale predictions of this nature. As illustrated for the Adenoviridae-human PPI example data (Fig. 4), random numbers for the Y2H case study will be inflated by a large over-prediction of kinase domain-mediated DMIs, as well as other domains with a specificity of interaction not captured at the level of Pfam definitions. Users may wish to screen out promiscuous domains and/or motifs if low stringency approaches are required to get sufficient DMI numbers.

## Using SLiMEnrich to assess enrichment of different PPI types

Although the focus of SLiMEnrich is on DMIs, the approach is flexible and can be easily adapted to other PPI types. Direct analogues of DMIs can be studied by replacing the motifs with a different interaction feature, e.g., replacing motifs with domains to investigate enrichment of Domain-Domain Interactions (DDIs). More simply, SLiMEnrich could be used to study the overlap between two different PPI datasets, accounting for the connectedness of the proteins involved, by replacing the known ELM interactions with any source of pairwise PPIs. Although the PPI data for the case study was made symmetrical, the asymmetrical handling of the PPI data by SLiMEnrich would even allow intra-dataset comparisons, such as examining the overlap between PPIs when proteins are baits versus preys in a Y2H or pulldown experiment.

## CONCLUSION

There are many data- and method-specific factors that will determine whether protein–protein interaction (PPI) data are useful for short linear motif (SLiM) prediction. The presence of real domain-motif interactions (DMIs) is a baseline requirement that is generally assumed but rarely tested. SLiMEnrich is an open source R application that will identify known or predicted DMIs in PPI data and estimate how well that PPI data are capturing DMIs compared to randomised PPIs. This estimate is useful for identifying suitable PPI data for *de novo* SLiM prediction. SLiMEnrich statistics also estimate the confidence in individual DMI predictions, enabling assessment of methods that aim to improve the specificity of DMI predictions by filtering SLiM predictions and/or PPI data.

Users can run SLiMEnrich online (http://shiny.slimsuite.unsw.edu.au/SLiMEnrich/) or download the code for local use (https://github.com/slimsuite/SLiMEnrich).

## ACKNOWLEDGEMENTS

The authors would like to thank the UNSW Science Faculty Computing unit, especially Adrian Plummer, for assistance in setting up the Shiny server. We thank Norman Davey for helpful comments on a draft of the manuscript. We also thank Benjamin Lang and two anonymous reviewers for insightful comments and suggestions on the original manuscript.

### Funding

This work was supported by the University of New South Wales through a University International Postgraduate Award to Sobia Idrees. The funders had no role in study design, data collection and analysis, decision to publish, or preparation of the manuscript.

### Grant Disclosures

The following grant information was disclosed by the authors:
University International Postgraduate Award to Sobia Idrees.

### Competing Interests

The authors declare there are no competing interests.

### Author Contributions

- Sobia Idrees performed the experiments, analyzed the data, contributed analysis tools, prepared figures and/or tables, authored or reviewed drafts of the paper, approved the final draft, developed the GitHub repository, and contributed additional documentation in project Wiki.
- Åsa Pérez-Bercoff analyzed the data, authored or reviewed drafts of the paper, approved the final draft, and contributed additional documentation in project Wiki.
- Richard J. Edwards conceived and designed the experiments, analyzed the data, contributed analysis tools, prepared figures and/or tables, authored or reviewed drafts of the paper, approved the final draft, developed the GitHub repository, and contributed additional documentation in project Wiki.

### Data Availability

SLiMEnrich code and example data: https://github.com/slimsuite/SLiMEnrich
SobiaIdrees, EdwardsLab, & asapb. (2018, July 25). slimsuite/SLiMEnrich: SLiMEnrich v1.5.1 (Version v1.5.1). Zenodo. http://doi.org/10.5281/zenodo.1320982
Edwards, Richard. 2018. "SLiMEnrich Y2H Case Study." OSF. July 25. doi: 10.17605/OSF.IO/VDXEN.

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
