# Peer review of "SLiM-Enrich: computational assessment of protein–protein interaction data as a source of domain-motif interactions"

_PeerJ, doi:10.7717/peerj.5858_

## Round 0.1 · original submission · Major Revisions

While the reviewers found your manuscript interesting and the presented resource potentially useful, they also raised significant criticisms. In particular, they requested the extension of the analyses to other datasets (e.g. HIPPIE, other Y2H datasets) and require further analyses to assess the accuracy of the tool.

Reviewer 1 ·

Basic reporting

Minor Comments:
1) There are a number of important reference missing from the paper, in particular, Neduva V, Russell RB (2006) and Gavin et al. (2006) (see the "validity of findings" section). Also, important are Pepsite (Petsalaki et al. (2009)), ADAN database (Encinar JA et al. (2009)), iELM (Weatheritt et al. (2012)), MiniMotif miner (Mi et al. (2012), and PIPER-FlexPepDock (Alam et al. (2017).) Plus review of Bilkstad and Ivarsson (2015).
2) The scripts used to create SLiMEnrich should be freely available.

Experimental design

No comment

Validity of the findings

Major Comments
1a) The authors they map “predicted SLiMs” to “any protein in the first PPI data column”. The authors have previously published programs such as SliMPrints and SLiMSearch that demonstrated the need to use conservation when predicting SLiMs. Based on the text, the authors are using only regular expression matches and a disorder filter. This is likely to lead to significant false positive rate. The authors need to demonstrate this does not impact their results (see point 1c).
1b) The authors identify SLiM-binding domains by identifying known interaction domains from the ELM-curated list in “any protein from the second PPI column”. However, many SLiMs have a strong preference for sub-groups of the Pfam domain families, as demonstrated by phage display and discussed in multiple papers from members of the ELM database A few prominent examples of this are SLiMs binding to the SH2, SH3, PDZ, and WD40 domain families. The authors need to demonstrate this does not impact their results (see point 1c).
1c) If the authors wish to use a weak stringency filter for SLiMEnrich, they need to demonstrate that the results are not affected by the stringency of filtering they use.

2) The authors describe a case study demonstrating the power of SLiMEnrich of Y2H data. The authors need to contrasts these results with Neduva and Russell in 2006, as well as in Gavin et al. (2006), that found “only about 1% of interaction in Y2H data rely on motif-mediated interaction”. The authors should also extend this analysis to compare multiple high-throughput PPI studies (e.g. from Y2H, tandem mass-spec, LUMIER, and MAPPIT), as well as older Y2H studies to the most recent iterations.

Additional comments

The authors introduce a computational pipeline called SliMEnrich to assess how well a given PPI dataset captures SLiM-mediated interactions. This is a useful approach to assess the enrichment of motifs within protein-protein interaction datasets. To the reviewer’s knowledge, the resource is unique and the website works nicely producing graphics that are useful and easy to interpret. However, the usefulness of the resource depends strongly on its accuracy, which is in not sufficiently assessed or demonstrated. Furthermore, Russell and colleagues did a similar analysis over 10 years ago, which the authors do not reference. Prior to publication, the authors need to address the handful of major comments below.

The website should provide the location of the motif and the domain in the output, as well as an assessment of accuracy.

Reviewer 2 ·

Basic reporting

- While the introduction explains the importance of SLiM and DMI prediction and
the motivation for developing SLiMEnrich, the authors fail to acknowledge a lot
of relevant work that has furthered our understanding of how SLiMs bind to
distinct recognition domains. Some of these papers exploit PPI data to predict
DMIs, so I disagree with the authors' statement that "the nature and complexity
of PPI data used for SLiM and DMI predictions has not been the focus of rigorous
study". Some of the papers I refer to are:
- https://doi.org/10.1186/1471-2105-12-225
- https://doi.org/10.1371/journal.pone.0025376
- https://doi.org/10.1093/bioinformatics/btu350
- https://doi.org/10.1371/journal.pcbi.1003881
- https://doi.org/10.1093/bioinformatics/btw045
- https://doi.org/10.1073/pnas.1518469113

- The authors must explain where SLiMEnrich stands compared to some of the
above-listed methods for DMI prediction, why it is better and list its
limitations.

Experimental design

- The authors claim that SLiMEnrich "assesses how well PPI data is capturing
DMIs and thus, by inference, how useful that data should be for SLiM discovery".
While it is clear how the former is achieved by comparison with a random
distribution of predicted DMIs, it is not clear how the pipeline can help with
SLiM discovery. The authors must explain this.

- I suggest that the authors label each step of the SLiMEnrich pipeline in
Fig. 1 (e.g. with letters or labels like "STEP 1") and refer to these sections
of the figure in the text. This will make it much easier to understand the
pipeline.

- One of the files required by SLiMEnrich is the SLiM predictions, which
associate proteins in the first column of the PPI file with short linear motifs.
The authors suggest the SLiMProb tool, developed by themselves, to generate this
mandatory file. What other tools are available for this task? Are their outputs
compatible with SLiMEnrich? What is the minimum set of columns required in a
SLiM-prediction file to be compatible with SLiMEnrich?

Validity of the findings

- The authors present a Case Study in which the well-known HI-II human protein
network is enriched with DMI data. In the Discussion, the authors argue that
this high-throughput PPI dataset is "indeed capturing real SLiM-mediated
interactions and [is] therefore suitable for de novo SLiM prediction". The
authors must clarify why the over-representation of DMIs in HI-II is unexpected.
Is it due to the low-affinity of SLiM-mediated interactions? If so, this could
be a good part to remind the reader of this fact.

- The authors conclude that "SLiMEnrich [...] is useful for identifying
suitable PPI data for de novo SLiM prediction". However, this is not supported
by the presented results. Even though the authors start the Discussion saying
that this is outside the scope of the paper, they must backup this conclusion
by applying their pipeline to different PPI data sources (e.g. a meta-database
like HIPPIE https://doi.org/10.1093/nar/gkw985, a Y2H dataset like HI-II and an
AP/MS dataset like the BioPlex https://www.nature.com/articles/nature22366) and
show that the cases in which DMI enrichments were poor lead to bad SLiM
predictions and vice versa.

·

Basic reporting

The article is well written and makes good use of the literature to provide sufficient context and background. The raw example data are downloadable at https://github.com/slimsuite/SLiMEnrich/tree/master/example, as referenced in the paper.

Self-contained: I would have liked to see the approach applied to other datasets to really address the challenge outlined in the abstract and introduction (comparing PPI detection methods). I think it would have been very easy to include one or two other protein-protein interaction datasets, and I don’t quite understand why the authors decided to stop where they did. A followup study is mentioned, however.

Figures:
- Figure 1 has some formatting issues and would greatly benefit from a simpler layout and use of colours. Some of the text is too small to be legible, the acronyms are not explained or obvious, and most of all the flow is not clear to me. Some arrows are missing heads. This figure does not currently benefit the paper as much as it could since it is unintelligible at first glance. What, conceptually, is the difference between “Tool/Application” and “Feature/Module”? It should be made clearer that SLiMProb is an external tool that provides input. What is an “ELMOcc”, an “EBD” and an “mProtein” to a reader trying to quickly grasp the workflow? An improved version would clearly show the input data sources and the outputs at a glance, without making use of acronyms. This would greatly improve the clarity of the paper.
- Figure 4 would greatly benefit from having intelligible protein names (e.g. gene symbols) and Pfam domain names rather than numeric identifiers — otherwise its usefulness is extremely limited. The same goes for the example data used in Figure 2 and on the website. I realise that no identifier mapping is done by the application, but it would be very helpful to have readable identifiers. A simple export from UniProt should provide this.

Finally, it would be useful if the Conclusions section spelled out the acronyms (lines 184-185).

Some minor points:
- Abstract, line 14 (“Despite of their significance”) -> “Despite their significance”
- Abstract, line 14 (“challenging to identify”): I think an explanation of such a statement is necessary, e.g. “due to their low affinity”
- Abstract, line 16 (“DMI are generally underrepresented”) should read “DMIs”

Experimental design

The application and analysis are very useful and admirably implemented for reproducibility. The approach is clear and straightforward and uses established methods. The analysis is open, well-documented, downloadable and additionally available as an interactive web implementation using Shiny.

The research question is extremely interesting. However, I am a bit surprised the authors decided to stop where they did, just short of actually comparing multiple PPI datasets, or using an ELM dataset containing only experimentally validated instances. However, a followup study is mentioned, which is great.

Validity of the findings

Using the example data on the website, the Histogram tab reports “False Discrovery Rate (FDR) is: 1.21”. It should not be above 1. This is not an actual “FDR” implementation and should be renamed (line 113). The definition is closer to a BLAST E-value. The text should also be amended where an “FDR” of 0.85 is reported (lines 174-175).

The conceptual issue here is that the mean 5,213 random DMIs are not necessarily all false, as the authors state in the introduction (the motivation being discovery of new domain-motif interactions, many of which do not show up in PPI datasets). I also find the very simplistic quantification of "true positives" in line 180 problematic. I am not sure the approach can support it.

A much better, and I would think more obvious measure would be to count how many of the ~14,000 Y2H interactions involve one of the 1,716 experimentally validated, true-positive human ELMs (http://elm.eu.org/instances/?q=&instance_logic=true+positive&taxon=Homo+sapiens&submit=submit&reset_form=Reset) and an appropriate ELM binding domain (http://elm.eu.org/interactiondomains).

Overall the analysis is clear and straightforward and I do find the findings believable, namely that Y2H is an appropriate method for domain-motif interaction detection, despite a high degree of noise.

---

## Round 0.2 · Minor Revisions

Although all three reviewers were very positive about the revised version of your manuscript, they still suggested some remaining minor revisions. Therefore, I would like give you another chance to consider their suggestions, before the manuscript is forwarded to the production team.

Congratulations on your work!

Reviewer 1 ·

Basic reporting

The discussion section should be limited to providing context to the results presented in the paper. At present, the discussion reads more like a review. I'd recommend making the discussion more succinct and saving discussion points for the author's future paper on PPI networks.

Experimental design

I am content with experimental design of the study

Validity of the findings

I am content the findings are valid

Additional comments

I am content with the authors' responses to my concerns. It is disappointing that they didn't include further analysis in the paper but I understand the logic.

Reviewer 2 ·

Basic reporting

The revised version of the article is well-written and easy to follow. The introduction explains the importance of SLiM and DMI prediction, includes the references suggested by me and the other reviewers and more clearly states the goal of SLiMEnrich.

Experimental design

The authors have made a much better version of Fig. 1 that makes it much easier to understand the proposed pipeline. In addition, they included a default set of SLiM predictions so that users can apply SLiMEnrich to their data out-of-the-box.

Validity of the findings

No comment.

·

Basic reporting

I still think the many acronyms used make the manuscript and tool unnecessarily cryptic. Terms such as "ELMc" really need to be explained, particularly in Table 1. I had to guess that this refers to ELM classes. Please define these in the text. "Motif-containing proteins" would not be unnecessarily long in place of "mProteins" either, I think.

Experimental design

I am very pleased that curated ELM instances were used now, and the low FDR estimate for these is reassuring to see.

Validity of the findings

This is still a bit difficult to assess when only looking at one PPI dataset, and I still believe that it would be very easy to test at least one or two additional ones. I do agree with the authors that the question of which experimental method and which specific PPI datasets best capture domain-motif interactions warrants an in-depth follow-up study, however, and I look forward to seeing it.

Additional comments

Thank you for the impressive amount of work involved in the improvements to the manuscript, the excellent new figures 1 and 2, and the improvements to SLiMEnrich itself. I very much look forward to reading your follow-up study.

---

## Round 0.3 · accepted · Accept

Thank you for taking the time to consider the reviewers' comments.

#